# Purified Distillation: Bridging Domain Shift and Category Gap in Incremental Object Detection

## ABSTRACT

Incremental Object Detection (IOD) simulates the dynamic data flow in real-world applications, which require detectors to learn new classes or adapt to domain shifts while retaining knowledge from previous tasks. Most existing IOD methods focus only on class incremental learning, assuming all data comes from the same domain. However, this is hardly achievable in practical applications, as images collected under different conditions often exhibit completely different characteristics, such as lighting, weather, style, etc. Class IOD methods suffer from severe performance degradation in these scenarios with domain shifts. To bridge domain shifts and category gaps in IOD, we propose Purified Distillation (PD), where we use a set of trainable queries to transfer the teacher's attention on old tasks to the student and adopt the gradient reversal layer to guide the student to learn the teacher's feature space structure from a micro perspective. This strategy further explores the features extracted by the teacher during incremental learning, which has not been extensively studied in previous works. Meanwhile, PD combines classification confidence with localization confidence to purify the most meaningful output nodes, so that the student model inherits a more comprehensive teacher knowledge. Extensive experiments across various IOD settings on six widely used datasets show that PD significantly outperforms state-of-the-art methods. Even after five steps of incremental learning, our method can preserve 60.6% mAP on the first task, while compared methods can only maintain up to 55.9%.

## CCS CONCEPTS

• **Computing methodologies → Object detection**.

## KEYWORDS

Incremental Learning, Object Detection, Catastrophic Forgetting

## 1 INTRODUCTION

Benefiting from the development of deep learning, object detection has achieved significant progress [10, 15, 37]. However, with the constant expansion of datasets and increasing number of categories, traditional training paradigms for object detection are facing new challenges [21], primarily in adapting to new domains [25] and categories [5, 11]. The simplest solution is to collect new data and retrain the model with both new and old data, but retraining a model from

*ACM MM, 2024, Melbourne, Australia*
© 2024 Copyright held by the owner/author(s). Publication rights licensed to ACM.
ACM ISBN 978-x-xxxx-xxxx-x/YY/MM
https://doi.org/10.1145/nnnnnnn.nnnnnnn

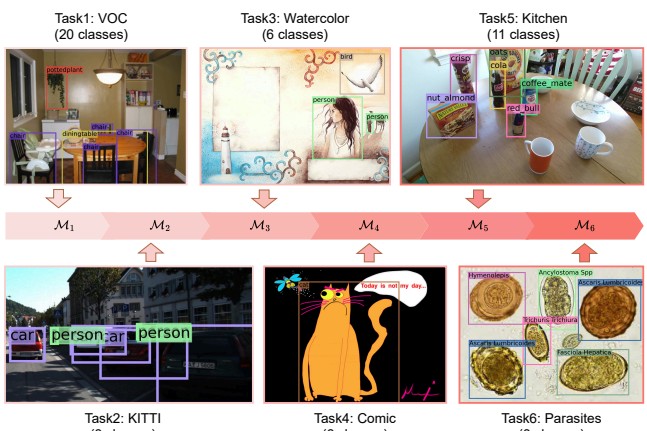

Figure 1: Illustration of incremental object detection with both domain shifts and category gaps. At time $T$, the incremental learned model $\mathcal{M}_T$ is expected to detect objects from any task $t$, where $t \leq T$.

scratch can be costly or impractical if the old data is no longer accessible. This necessitates engineers to finetune models exclusively with new data, while maintaining the knowledge from previous data, i.e., to avoid catastrophic forgetting [12].

Incremental Object Detection (IOD) [24, 26, 33] formalizes this problem, where the model continually accumulates new knowledge without forgetting previous knowledge. Existing works [5, 11, 13, 26, 27, 31] predominantly focused on class incremental learning, assuming that data for both new and old tasks comes from the same domain, and new categories are introduced in the new task. Nevertheless, this assumption proves overly restrictive in real-world applications, where models are commonly deployed across diverse contexts [21]. This implies that data for both new and old tasks often originates from different domains, thereby presenting a more challenging and less explored problem, incremental object detection with domain shift and category gap. As shown in Fig. 1, the detector encounters a series of tasks, each with potentially different categories and domains. After incremental learning, the detector should be able to remember all previously seen classes and domains. Liu *et al.* [21] initiated exploration into this problem and proposed Attentive Feature Distillation (AFD) as an attempt to alleviate forgetting. The efficacy of AFD largely relies on exemplars preserved from old data, but accessing old data can be impractical, and an excessive number of exemplars increases training costs. Therefore, further exploring the informative knowledge held by teacher models, we propose a novel distillation approach that requires no exemplars from previous tasks to bridge domain shifts and category gaps in IOD.

IOD methods typically utilize features and outputs distillation [26, 27] to prevent catastrophic forgetting, where the model trained on the

old task serves as a teacher. Features extracted by the teacher prove to focus on areas beneficial for old classes and domains, hence some works [13, 21, 27] minimize the differences ($L_2$ or $L_1$ loss) between the teacher's and student's features to help preserve the student's perception of old tasks. However the teacher focuses exclusively on the old task, the response to the new task is limited. Such feature distillation approaches can confuse the student model during the learning of new tasks. On the other hand, guiding the student model using the teacher's outputs is also a common practice [5, 27, 30]. The outputs are often noisy and exhibit a significant imbalance between positive and negative samples [5], so it is necessary to select the most meaningful nodes to contribute distillation, typically based on a top-$k$ strategy [30] or by setting a threshold [5]. Nevertheless, we notice that these selection strategies introduce some low-quality nodes, such as those with high classification scores but low confidence in bounding box positions, as detailed in the appendix.

To address the above issues, we delve into the analysis of features and outputs extracted by the teacher model, proposing Purified Distillation (PD). PD employs three distillation strategies: Multi-scale Cross Attention Distillation (MCAD), Feature Space Distillation (FSD), and Entropy Guided Output Distillation (EGOD). **MCAD**: We use trainable parameters as queries to transform the multi-scale feature maps extracted from both the teacher and student into attention maps [35]. Subsequently, by minimizing differences between the attention maps of the teacher and student, these queries adaptively purify regions preserving previous knowledge, providing flexibility for balanced learning between current and previous tasks. **FSD**: We notice that the trained model possesses a unique feature space, which reflects the abstract characteristics of training data but has been overlooked by previous methods. Based on this observation, we propose to transfer the structural properties of the teacher's feature space from a microscopic perspective. Specifically, we design a multi-scale feature discriminator and adopt the Gradient Reversal Layer (GRL) [6] to encourage the student to learn a feature space that simultaneously exhibits characteristics of both new and old tasks. In this way, the student's feature space can be seen as an approximation of full dataset training. **EGOD**: Inspired by modeling a bounding box as a discrete distribution [16], we propose entropy guided output distillation, which employs an entropy guided selection strategy to choose the most important output nodes. This strategy treats the entropy of the distribution as the location confidence and integrates classification confidence to select the top-$k$ bounding boxes to teach the student model. Extensive qualitative and quantitative experiments demonstrate that PD can preserve more knowledge of previous tasks and avoid catastrophic forgetting. Since PD does not require any modifications to the detector's structure, the detector can still maintain its efficiency during inference.

In summary, our contributions are as follows:

- To our knowledge, the proposed purified distillation is the first work to achieve incremental object detection with both domain shift and category gap, and eliminates dependence on exemplars from old tasks.
- By leveraging learnable parameters to query feature maps, we propose multi-scale cross attention distillation to enable the model to adaptively query information that is more meaningful for old knowledge.

- We devise a multi-scale feature discriminator and utilize a gradient reversal layer to distill the structure of feature space, so that the student can learn feature characteristics of both new and old tasks.
- We develop the entropy guided selection strategy to purify more representative nodes from outputs, which considers both the location confidence and classification scores to reduce redundant information.
- We construct multiple incremental object detection scenarios across five widely used datasets, and extensive experiments demonstrated the advantages of our method in mitigating catastrophic forgetting.

## 2 RELATED WORKS

### 2.1 Object Detection

Object detection has been extensively studied. Transformer-based detectors [1, 14, 19, 37, 40] commonly train a set of queries to locate objects within images, whereas suffering from significant computation. CNN-based detectors can be categorized into two-stage and one-stage detectors. As the most classic two-stage detector, Faster R-CNN [30] utilizes a region proposal network to generate proposals, followed by separate branches for computing classification scores and location offsets. Due to the complex forward procedure, two-stage detectors are not applicable for real-time scenarios. While one-stage detectors [20, 32, 38] directly feed multi-level feature maps into detection heads to predict classification scores and locations. Furthermore, Li *et al.* [16] proposed Generalized Focal Loss (GFL), modeling the location output in an integral form to reduce model ambiguity and uncertainty. The latest work in the YOLO series [28, 29, 36], YOLOv8 [10], incorporates GFL and task-aligned sample assigner [4], achieving state-of-the-art in both efficiency and accuracy.

### 2.2 Incremental learning

Recently, incremental learning [24, 26, 33] has received much attention, which requires the model trained on the new task to retain knowledge from the old task. To avoid catastrophic forgetting, knowledge distillation [17] is widely used to transfer the teacher's knowledge to the student. Shmelkov *et al.* [31] pioneered incremental object detection by leveraging distillation losses, effectively maintaining the memory of old tasks. Afterwards, Li *et al.* [13] and Peng *et al.* [26] further proposed distillation methods for RetinaNet [18] and Faster R-CNN [30], respectively. SID [27] characterized the relation between instances, proposing selective and inter-related distillation for anchor-free detectors [32, 38]. While Liu *et al.* [22] proposed CL-DETR, a transformer-based IOD, by utilizing knowledge distillation and exemplar replay. Considering the inadequacy of existing methods in exploring responses from detection heads, Feng *et al.* [5] introduced elastic response distillation, selectively distilling nodes from output responses.

Despite their success in class incremental object detection, the problem with both domain shift and category gap has been overlooked. Liu *et al.* [21] made the first work to explore this problem and proposed Attentive Feature Distillation (AFD) composed of top-down and bottom-up feature distillations. AFD relies heavily on a set of exemplars from old tasks, but this could be impractical

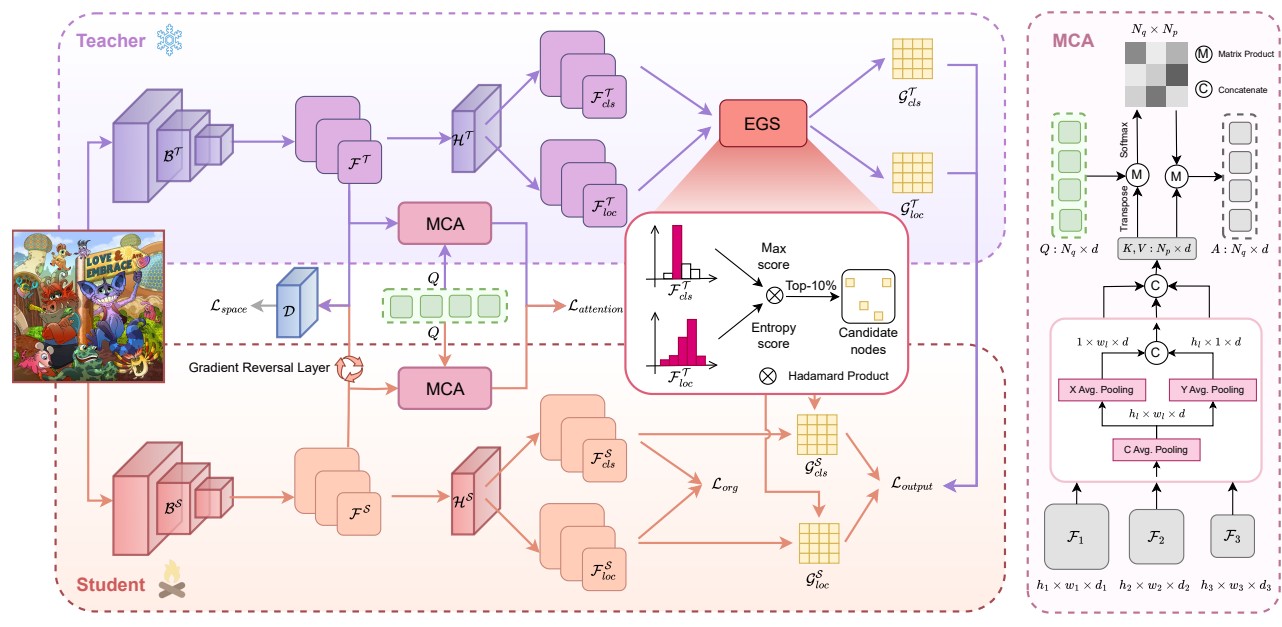

**Figure 2: Overall structure of proposed Purified Distillation. The teacher model trained on the old task serves as a knowledge repository, while the student model is trained on the new task, starting with the initial weights of the teacher. Knowledge from the old task is imparted to the student model through multi-scale cross attention distillation, feature space distillation, and entropy guided output distillation.**

in real-world applications. Thus, we propose a more feasible and realistic distillation method that does not require any exemplars, alleviating catastrophic forgetting in incremental object detection with both domain shift and category gap.

## 3 METHOD

### 3.1 Overall Structure

We construct PD on the single-stage detector which can be divided into two key components: the backbone that extracts features, and the detection head that predicts the category probability and location offsets. Considering the two components, we propose multi-scale cross attention distillation and feature space distillation to preserve knowledge from old tasks in the latent space, and the entropy guided selection strategy to transfer the classification and location information of the valuable output nodes.

The architecture of our method is depicted in Fig. 2. The teacher detector trained on the old task provides initial weights of the student model, which will be incrementally trained on the new task. During incremental training, the teacher is frozen, and only the student is trainable. The backbones for the teacher and student, denoted as $\mathcal{B}^{\mathcal{T}}$ and $\mathcal{B}^{\mathcal{S}}$, extract multi-scale features $\mathcal{F}^{\mathcal{T}}$ and $\mathcal{F}^{\mathcal{S}}$ respectively. We fuse the multi-scale feature maps, and construct trainable parameters $Q$ to adaptively query features that are crucial for maintaining previous knowledge. Moreover, a feature discriminator is responsible for distinguishing whether feature maps are extracted by the student. We employ a gradient reversal layer to reverse the gradient backward from the discriminator, ensuring the transfer of the structural characteristic of feature space. Feature map $\mathcal{F}$ is subsequently passed through the detection head $\mathcal{H}$ to predict classification scores $\mathcal{F}_{cls}$

and location offsets $\mathcal{F}_{loc}$. To purify output nodes, we propose the Entropy Guided Selection (EGS) strategy, which selectively transfers semantic and location knowledge from teacher to student.

Overall, the loss function of our incremental learning method is as follows:

$$\mathcal{L} = \mathcal{L}_{org} + \lambda\mathcal{L}_{pd}, \tag{1}$$

where $\mathcal{L}_{org}$ is the standard loss of the single-stage detector for learning the new task, typically composed of classification and localization losses, and $\mathcal{L}_{pd}$ is the distillation loss of the proposed PD. The parameter $\lambda$ balances the distillation and standard training, empirically set to 10.

### 3.2 Multi-scale Cross Attention Distillation

To transfer the knowledge to the student in the latent space and avoid adversely affecting the current task, we utilize a group of trainable parameters as queries to adaptively focus the feature most crucial to the old task. The features $\mathcal{F}$ extracted by the backbone include $L$ levels ($L = 3$ typically) of multi-scale features, and the features at the $l$-th level can be denoted as $\mathcal{F}_l \in \mathbb{R}^{w_l \times h_l \times d_l}$, where $w_l$, $h_l$, and $d_l$ are the width, height, and number of channels of the feature map, respectively. The queries can be considered as a matrix $Q \in \mathbb{R}^{n_q \times d}$, where $n_q$ and $d$ are the number of queries and dimensions, respectively. Without loss of generality, we set $d$ to the dimension of the feature map at the deepest level.

Taking into account that the channels of feature maps vary across levels, and larger feature maps can lead to significant computational costs, we devise Multi-scale Cross Attention (MCA) to query the attention map by down-sampling and integrating feature maps at

different levels. As illustrated in the right of Fig. 2, using channel-wise average pooling, the MCA first aligns the channels of feature maps to $d$ on each level. Then the features are concentrated along the horizontal and vertical directions, so that the key and value matrices are obtained by concatenating the pooled features:

$$\mathcal{F}_{p,l} = Cat([\mathcal{P}_x(\mathcal{P}_c(\mathcal{F}_l)), \mathcal{P}_y(\mathcal{P}_c(\mathcal{F}_l))]), \quad (2)$$

$$K = V = Cat([\mathcal{F}_{p,1}, \cdots, \mathcal{F}_{p,L}]) \in \mathbb{R}^{n_p \times d}, \quad (3)$$

where $\mathcal{P}_c$, $\mathcal{P}_x$, and $\mathcal{P}_y$ are the average pooling layer over the channel, x-axis, and y-axis, respectively; $Cat$ is the concatenation operation; and $n_p = \sum_{l=1}^{L} w_l + h_l$. Finally, given the softmax function $\tau$, the attention map is:

$$A = \tau \left( \frac{Q \times K^\top}{\sqrt{d}} \right) \times V. \quad (4)$$

As trainable parameters, queries $Q$ have considerable flexibility to adapt and learn the intricate dependencies among features, thus the attention maps $A$ queried by $Q$ can gradually capture knowledge critical to old tasks during training. By compelling the student's attention map $A^{\mathcal{S}}$ to mimic the teacher's $A^{\mathcal{T}}$, the student can retain memory about old tasks. Hence, the attention distillation loss $\mathcal{L}_{attention}$ is defined as follows:

$$\mathcal{L}_{attention} = \frac{1}{L} \sum_{l=1}^{L} \mathcal{L}_{MSE} \left( A_l^{\mathcal{S}}, A_l^{\mathcal{T}} \right), \quad (5)$$

where $\mathcal{L}_{MSE}$ is the mean square error loss.

### 3.3 Feature Space Distillation

We observed that models trained on different datasets possess unique feature spaces (detailed in Sec. 4.1), meaning that the feature vectors extracted by a model are distributed in a specific region of the latent space. Therefore, we argue that the teacher's feature space can reflect its knowledge about the abstract style of data in the old task, and introduce Feature Space Distillation (FSD) to transfer this property to the student model. Because of the challenge of quantitatively representing the feature space, we investigate utilizing individual feature maps to transfer the structure of the feature space. Directly applying an $L_2$ loss, however, would lead to conflicts between new and old knowledge, thus we innovatively introduce a Gradient Reversal Layer (GRL) [6] into incremental object detection to encourage the student to extract more "old-task-style" feature maps, instead of requiring strict consistency between student and teacher. By trying to confuse the discriminator, the student model can learn a feature space that can represent both new and old task characteristics, closer to that of the full dataset training.

Existing discriminators are mostly based on the patch discriminator [39], which accepts a single image as input and outputs the probability of whether each patch is synthesized, but is unsuitable for evaluating multi-scale feature maps. Feature maps at different scales represent objects and backgrounds from different perspectives, and isolated processing of different scales of feature maps overlooks their intrinsic connections. Therefore, we construct an efficient multi-scale feature discriminator, which comprehensively leverages the semantic information of feature maps at different levels, as depicted in Fig. 3.

We use convolutional layers with different strides to align the feature maps to the same dimension and fully connected layers to

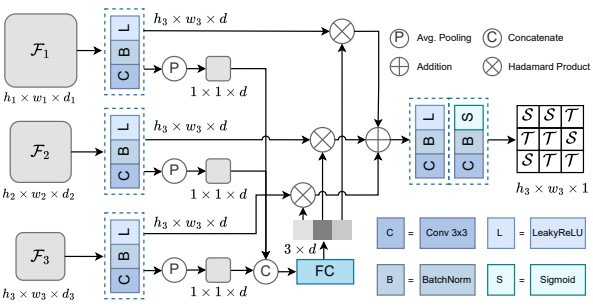

**Figure 3: The architecture of the multi-scale feature discriminator. The output is a logit matrix, representing whether each patch is extracted by the student ($\mathcal{S}$) or teacher ($\mathcal{T}$).**

reweight feature maps at different scales. The reweighted feature maps are then added together and fed into two convolutional layers to predict the discrimination results. The output of the discriminator is a logits matrix $O_l \in \mathbb{R}^{w_l \times h_l}$, which determines whether a patch belongs to the student or teacher. The teacher feature maps $\mathcal{F}_l^{\mathcal{T}}$ are directly fed into the discriminator, while the student features $\mathcal{F}_l^{\mathcal{S}}$ require passing through the GRL before inputting the discriminator $\mathcal{D}$, detailed as follows:

$$O_l^{\mathcal{T}} = \mathcal{D} \left( \mathcal{F}_l^{\mathcal{T}} \right), \quad (6)$$

$$O_l^{\mathcal{S}} = \mathcal{D} \left( \mathcal{R} \left( \mathcal{F}_l^{\mathcal{S}} \right) \right), \quad (7)$$

where $\mathcal{R}$ is the GRL. During the forward propagation, $\mathcal{R}$ does not modify its input, whereas during the backward propagation, it reverses the sign of the gradient. $[P_l]_{i,j}$ represents the confidence that the patch belongs to the student, and the corresponding target is defined as $[Y_l^{\mathcal{S}}]_{i,j} = 1$ and $[Y_l^{\mathcal{T}}]_{i,j} = 0$, thus the feature space distillation loss $\mathcal{L}_{space}$ is defined as:

$$\mathcal{L}_{space} = \frac{1}{L} \sum_{l=1}^{L} \mathcal{L}_{BCE} \left( O_l^{\mathcal{T}}, Y_l^{\mathcal{T}} \right) + \mathcal{L}_{BCE} \left( O_l^{\mathcal{S}}, Y_l^{\mathcal{S}} \right), \quad (8)$$

where $\mathcal{L}_{BCE}$ is the binary cross entropy loss. As training progresses, the discriminator tries to discern whether the input feature is extracted from the teacher or student model, while the student aims to extract features that confuse the discriminator, aligning its features more similar to those of the teacher.

### 3.4 Entropy Guided Output Distillation

The outputs from the detection head represent the semantic and location knowledge held by the model but contain a considerable amount of redundant information, therefore, we define the Entropy Guided Selection (EGS) to transfer classification and location information from the teacher to the student selectively. Specifically, the $l$-th detection head $\mathcal{H}_l$ takes the feature maps $\mathcal{F}_l$ as inputs, and outputs two sets of predictions: classification scores $\mathcal{F}_{cls,l} \in \mathbb{R}^{w_l \times h_l \times n_c}$ and location offsets $\mathcal{F}_{loc,l} \in \mathbb{R}^{w_l \times h_l \times 4 \times n_{loc}}$, where $n_{loc}$ is the number of sampling points for the distribution, typically set to 16; $n_c$ is the number of classes, and $n_c^{\mathcal{T}} \leq n_c^{\mathcal{S}}$.

For the teacher's classification scores $\mathcal{F}_{cls,l}^{\mathcal{T}}$, we select the maximum over the class dimension to obtain the class selection scores $S_{cls,l} \in \mathbb{R}^{w_l \times h_l}$:

$$[S_{cls,l}]_{i,j} = \max\left\{\sigma\left(\left[\mathcal{F}_{cls,l}^{\mathcal{T}}\right]_{i,j,k}\right) | k = 1, 2, \cdots, n_c^{\mathcal{T}}\right\}, \quad (9)$$

where $\sigma$ is the sigmoid function.

For the teacher's location offsets $\mathcal{F}_{loc,l}^{\mathcal{T}}$, we first apply the softmax function to transform the location of any edge into a discrete probability distribution. Then, we can calculate the entropy of each edge to measure the detector's location confidence as follows:

$$[E_l]_{i,j,k} = -\sum_{x=1}^{n_{loc}} [C_l^{\mathcal{T}}]_{i,j,k,x} \log [C_l^{\mathcal{T}}]_{i,j,k,x}, \quad (10)$$

where $[C_l^{\mathcal{T}}]_{i,j,k} \in \mathbb{R}^{n_{loc}}$ and $[E_l^{\mathcal{T}}]_{i,j,k} \in \mathbb{R}$ is the distribution and the entropy of the $k$-th edge of bounding box $(i, j)$ respectively.

In a discrete probability distribution, the maximum entropy occurs when all possible outcomes have equal probability, yielding an entropy of $\log n_{loc}$; the minimum entropy occurs when one outcome has a probability of 1 while others are 0, resulting in zero entropy. Thus we can normalize the entropy to the range of $[0, 1]$:

$$E_l' = \frac{\log n_{loc} - E_l}{\log n_{loc}}. \quad (11)$$

A larger normalized entropy $E_l'$ indicates the detector has higher confidence in localizing that edge, and vice versa. For bounding box $(i, j)$, we take the minimum normalized entropy of the four edges as the location selection score $[S_{loc,l}]_{i,j}$:

$$[S_{loc,l}]_{i,j} = \min\{[E_l']_{i,j,k} | k = 1, 2, 3, 4\}. \quad (12)$$

By multiplying the class and location selection scores, we obtain the selection score for the $l$-th level:

$$S_l = S_{cls,l} \otimes S_{loc,l}, \quad (13)$$

where $\otimes$ is the Hadamard product.

We select the top-$k$ nodes with the highest selection scores, denoted as $\mathcal{G}_{cls,l}$ and $\mathcal{G}_{loc,l}$. The output distillation loss $\mathcal{L}_{output}$ is defined as:

$$\mathcal{L}_{output} = \frac{1}{L}\sum_{l=1}^{L}\mathcal{L}_{KL}\left(\sigma\left(\mathcal{G}_{cls,l}^{\mathcal{S}}/t\right), \sigma\left(\mathcal{G}_{cls,l}^{\mathcal{T}}/t\right)\right) +$$
$$\mathcal{L}_{KL}\left(\tau\left(\mathcal{G}_{loc,l}^{\mathcal{S}}/t\right), \ \tau\left(\mathcal{G}_{loc,l}^{\mathcal{T}}/t\right)\right), \quad (14)$$

where $\mathcal{L}_{KL}$ is the KL divergence loss; $\sigma$ and $\tau$ are sigmoid and softmax function respectively; $t$ is the temperature, typically set to 2.

### 3.5 Total Distillation Loss

Given the attention distillation loss, the feature space distillation loss, and the entropy guided output distillation loss, we formalize the total distillation loss $\mathcal{L}_{pd}$ as follows:

$$\mathcal{L}_{pd} = \mathcal{L}_{attention} + \mathcal{L}_{space} + \mathcal{L}_{output}. \quad (15)$$

By purifying the teacher's knowledge from various perspectives, $\mathcal{L}_{pd}$ facilitates the transfer of the most meaningful knowledge from the teacher to the student, and prevents interference from previous knowledge during learning new tasks.

**Table 1: Datasets and categories. The overlapping categories are highlighted in the same colors.**

| Datasets | Categories |
|---|---|
| VOC [3] | bicycle, car, person, bird, cat, dog, aeroplane, boat, bottle, bus, chair, cow, horse, motorbike, train, diningtable, pottedplant, sheep, sofa, tvmonitor |
| Comic [9] | bicycle, car, person, bird, cat, dog |
| Watercolor [9] | bicycle, car, person, bird, cat, dog |
| KITTI [7] | bicycle, car, person |
| Kitchen [8] | coffee mate, cola, crisp, nut bar, oatmeal squares, oats, popcorn, red bull, rice, sauce, dish soap |
| Parasites [2] | ancylostoma spp, taenia sp, trichuris trichiura, ascaris lumbricoides, hymenolepis, schistosoma, fasciola hepatica, enterobius vermicularis |

## 4 EXPERIMENTS

*Datasets and Metrics.* To validate the generalizability of our method, we employ six commonly used datasets: PASCAL VOC [3], Watercolor [9], Comic [9], KITTI [7], Kitchen [8], and Parasites [2], detailed in Tab. 1. To quantify the detection performance, we use the mean Average Precision at the Intersection over Union (IoU) threshold of 0.5 (mAP@0.5).

*Implementation details.* Based on YOLOv8, we implement our PD and other methods, including vanilla finetuning, LwF [17], Faster ILOD [26], SID [27], and ERD [5]. We train the model on the old task from the official weights [10], subsequently, this model is treated as the common teacher of different IOD methods. In each task, the model is trained for 20 epochs with a batch size of 16, a learning rate of 1e-4, and the AdamW optimizer [23], while other training parameters are the same as the original YOLOv8. All experiments are conducted on a single RTX 3090 GPU. For hyperparameters, we set $k = 10\%$ and $n_q = 512$ whose sensitive analyses are detailed in the appendix.

### 4.1 Visualization of the feature space

To visualize feature spaces, we train a model on each dataset, then extract feature maps of samples from all six datasets and plot them in Fig. 5a using t-SNE [34] for dimensionality reduction. We can see that even samples not in the training dataset have feature maps distributed near the training samples, forming a unique region. This suggests that the structure of feature space represents the model's certain knowledge about the characteristics of the training dataset, which has not been explored by existing methods. To verify whether Feature Space Distillation (FSD) can transfer knowledge of feature space structure, we remove FSD from our method and compare it with different methods in the scenario (Kitchen → KITTI), as shown in Fig. 5b. Compared to other methods (including PD without FSD), PD has the highest overlap with joint training in feature space, indicating that FSD effectively guides the student to extract features that exhibit both new and old task characteristics.

**(a). Finetuning**

**(b). LwF**

**(c). Faster ILOD**

**(d). SID**

**(e). ERD**

**(f). Ours**

| VOC → Comic | KITTI → Kitchen | Kitchen → Parasites | Parasites → Comic | Watercolor → Comic | VOC[:10] → VOC[10:] |

**Figure 4: The qualitative results under three scenarios for different methods: (a) Vanilla Finetuning, (b) LwF [17], (c) Faster ILOD [26], (d) SID [27], (e) ERD [5], and (f) Ours. We provide the predictions on old tasks after incremental learning on new tasks. We can see that competitive methods exhibit significant catastrophic forgetting, manifested as the inability to detect objects from old tasks or misclassification into other categories. In contrast, our method excels at preserving knowledge of old tasks.**

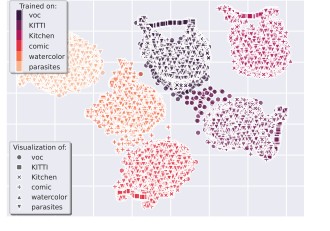

**(a) Different datasets**

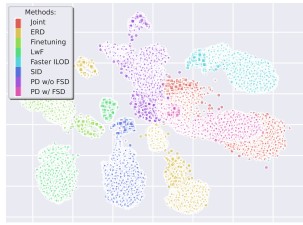

**(b) Different IOD methods**

**Figure 5: The visualization of feature spaces. (a) Different datasets: The models are trained on different datasets and visualize feature maps in all datasets. (b) Different IOD methods: The models are incrementally learned with different methods under the scenario (Kitchen → KITTI) and visualize feature maps in all datasets. "Joint" is training with both new and old datasets, and "PD w/o FSD" is our method without FSD.**

## 4.2 One-step Incremental Object Detection

One-step incremental object detection can be categorized into three scenarios: "Domain Shift", "Category Gap", and "Domain Shift + Category Gap". In detail, the dataset pairs (Kitchen, KITTI), (Comic, Parasites), (Kitchen, Parasites), and (VOC, Comic) differ in both domains and categories across old and new tasks; the dataset pair (Watercolor, Comic) only differs in domains with the same categories; and the dataset pair (VOC[:10], VOC[10:]) only differs in categories with the same domain, where VOC[:10] and VOC[10:] denote the sub-dataset that contain the first and last ten classes of VOC. Tab. 2, Tab. 3, and Tab. 4 illustrate the experimental results of our method compared to others under scenarios with both category gap and domain shift, only category gap, and only domain shift, respectively. It can be observed that models finetuned achieve high accuracy on new tasks but fail to retain the knowledge of old tasks, resulting in catastrophic forgetting. Although LwF is a classic method in incremental classification, it proves inadequate in IOD and tends to forget old tasks. SID shows some improvement but

**Table 2: The incremental learning results (mAP@0.5) under scenarios with both category gap and domain shift. "Teacher" denotes the model trained on the old task. The values in parentheses represent the accuracy loss on the old task after incremental learning, providing a quantification of catastrophic forgetting.**

| Methods | Kitchen → KITTI | | | KITTI → Kitchen | | | Comic → Parasites | | | Parasites → Comic | | |
|---|---|---|---|---|---|---|---|---|---|---|---|---|
| | Old | New | Avg. | Old | New | Avg. | Old | New | Avg. | Old | New | Avg. |
| Teacher | 94.73 | - | - | 74.83 | - | - | 56.44 | - | - | 81.89 | - | - |
| Finetuning | 0.00 (-94.73) | 72.08 | 36.04 | 0.00 (-74.83) | 93.30 | 46.65 | 0.00 (-56.44) | 80.20 | 40.10 | 0.00 (-81.89) | 55.32 | 40.94 |
| LwF [17] | 60.52 (-34.21) | 68.03 | 64.27 | 54.36 (-20.47) | 87.22 | 70.79 | 18.95 (-37.49) | 76.76 | 47.85 | 24.40 (-57.49) | 42.08 | 33.24 |
| Faster ILOD [26] | 85.05 (-9.68) | 68.07 | 76.56 | 68.63 (-6.20) | 92.15 | 80.39 | 48.31 (-8.13) | 80.00 | 64.15 | 71.16 (-10.73) | 54.91 | 63.03 |
| SID [27] | 78.58 (-16.15) | 64.07 | 71.32 | 57.49 (-17.34) | 88.29 | 72.89 | 32.57 (-23.87) | 71.23 | 51.90 | 65.03 (-16.86) | 48.69 | 56.86 |
| ERD [5] | 83.40 (-11.33) | 70.51 | 76.95 | 64.23 (-10.60) | 92.07 | 78.15 | 49.20 (-7.24) | 79.88 | 64.54 | 72.46 (-9.43) | 56.05 | 64.25 |
| **Ours** | 89.70 (-5.03) | 71.22 | **80.46** | 73.18 (-1.65) | 93.59 | **83.38** | 52.39 (-4.05) | 81.07 | **66.73** | 76.58 (-5.31) | 55.77 | **66.17** |

| Methods | Kitchen → Parasites | | | Parasites → Kitchen | | | VOC → Comic | | | Comic → VOC | | |
|---|---|---|---|---|---|---|---|---|---|---|---|---|
| | Old | New | Avg. | Old | New | Avg. | Old | New | Avg. | Old | New | Avg. |
| Teacher | 94.73 | - | - | 81.89 | - | - | 87.76 | - | - | 56.44 | - | - |
| Finetuning | 0.00 (-94.73) | 81.55 | 40.77 | 0.00 (-81.89) | 95.01 | 47.50 | 59.06 (-28.70) | 56.01 | 57.53 | 41.70 (-14.74) | 88.39 | 65.04 |
| LwF [17] | 30.56 (-64.17) | 71.10 | 50.83 | 49.96 (-31.93) | 86.58 | 68.27 | 66.34 (-21.42) | 55.35 | 60.84 | 43.69 (-12.75) | 85.02 | 64.35 |
| Faster ILOD [26] | 87.80 (-6.93) | 78.05 | 82.92 | 76.38 (-5.51) | 92.06 | 84.22 | 80.09 (-7.67) | 54.23 | 67.16 | 53.38 (-3.06) | 85.29 | 69.33 |
| SID [27] | 68.56 (-26.17) | 74.34 | 71.45 | 62.50 (-19.39) | 88.20 | 75.35 | 70.18 (-17.58) | 52.64 | 61.41 | 49.51 (-6.93) | 83.04 | 66.27 |
| ERD [5] | 85.01 (-9.72) | 78.49 | 81.75 | 75.04 (-6.85) | 91.65 | 83.34 | 76.71 (-11.05) | 54.39 | 65.55 | 52.97 (-3.47) | 88.02 | 70.49 |
| **Ours** | 90.35 (-4.38) | 80.09 | **85.22** | 78.62 (-3.27) | 92.44 | **85.53** | 83.15 (-4.61) | 55.15 | **69.15** | 55.82 (-0.62) | 87.59 | **71.70** |

**Table 3: The incremental learning results (mAP@0.5) under scenarios with domain shift only.**

| Method | Watercolor → Comic | | | Comic → Watercolor | | |
|---|---|---|---|---|---|---|
| | Old | New | Avg. | Old | New | Avg. |
| Teacher | 64.24 | - | - | 56.44 | - | - |
| Finetuning | 61.83 (-2.41) | 56.51 | 59.17 | 53.79 (-2.72) | 65.61 | 59.70 |
| LwF [17] | 65.44 (+1.20) | 55.49 | 60.46 | 57.72 (+1.28) | 64.26 | 60.99 |
| Faster ILOD [26] | 65.81 (+1.57) | 49.31 | 57.56 | 59.16 (+2.72) | 64.04 | 61.60 |
| SID [27] | 66.78 (+2.54) | 55.03 | 60.90 | 58.75 (+2.31) | 65.33 | 62.04 |
| ERD [5] | 66.54 (+2.30) | 51.56 | 59.05 | 58.88 (+2.44) | 65.60 | 62.24 |
| **Ours** | 68.25 (+4.01) | 56.60 | **62.42** | 61.03 (+4.59) | 66.24 | **63.63** |

**Table 4: The incremental learning results (mAP@0.5) under scenarios with category gap only.**

| Method | VOC[:10] → VOC[10:] | | | VOC[10:] → VOC[:10] | | |
|---|---|---|---|---|---|---|
| | Old | New | Avg. | Old | New | Avg. |
| Teacher | 81.86 | - | - | 86.14 | - | - |
| Finetuning | 0.00 (-81.86) | 78.88 | 39.44 | 0.00 (-86.14) | 88.49 | 44.24 |
| LwF [17] | 53.93 (-27.93) | 72.75 | 63.34 | 54.13 (-32.01) | 88.34 | 71.23 |
| Faster ILOD [26] | 77.84 (-4.02) | 64.09 | 70.96 | 84.10 (-2.04) | 90.45 | 87.27 |
| SID [27] | 74.99 (-6.87) | 72.58 | 73.78 | 83.39 (-2.75) | 90.32 | 86.85 |
| ERD [5] | 80.13 (-1.73) | 73.44 | 76.78 | 84.66 (-1.48) | 90.00 | 87.33 |
| **Ours** | 79.88 (-1.98) | 74.00 | **76.94** | 85.28 (-0.86) | 90.41 | **87.84** |

maintaining memory of old tasks has caused significant interference with learning new tasks. As the state-of-the-art method in class incremental object detection, ERD outperforms other methods in the "Category Gap" scenario but falls short in scenarios with domain shift. We also notice that despite being a relatively outdated method, Faster ILOD performs comparably to ERD in many scenarios, and even surpasses ERD. In comparison, our PD achieves up to 4.65% higher mAP compared to the state-of-the-art method in maintaining knowledge of old tasks. Moreover, we visualize the qualitative results of incremental learning in different scenarios in Fig. 4. It can be observed that even the most competitive method, ERD, inevitably exhibits forgetting of old tasks, incorrectly identifying objects as background or other categories, while our PD effectively detects objects from both new and old tasks.

## 4.3 Multi-step Incremental Object Detection

In real-world applications, detectors are often expected to incrementally learn multiple tasks. Therefore we construct a task sequence with six datasets (VOC → KITTI → Watercolor → Comic → Kitchen → Parasites), and report the detection performance on all previously learned datasets after each incremental step. The results of multi-step incremental object detection are visualized in mAP matrices, as shown in Fig. 6. We can tell that PD effectively preserves the knowledge of old tasks while adapting to new ones. After five steps of incremental learning, our method retains 60.6% mAP for the first task, whereas the least effective comparative method, LwF, manages to preserve only 10.1%. Even the best-performing comparative method, Faster ILOD, is only able to maintain 55.9% detection performance for the first task.

## 4.4 Ablation Experiments

In this section, we select the two most challenging scenarios: (Kitchen → KITTI) and (Comic → Parasites), and conduct detailed ablation experiments on three distillation losses. As shown in Tab. 5,

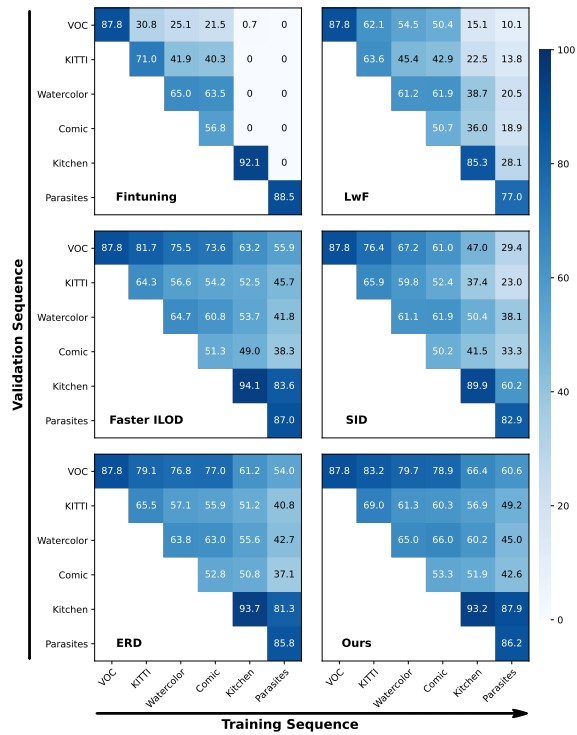

**Figure 6: The mAP matrices of different methods. The horizontal axis represents the dataset sequence for incremental learning, with a total of five incremental steps. After each incremental step, we validate the model on all seen tasks and report the metrics (mAP@0.5).**

**Table 5: Ablation experiments of proposed PD in three scenarios. The symbol × indicates that we set the weight of this loss term to 0, and ✓ indicates that we keep this loss.**

| #Exp. | $\mathcal{L}_{attention}$ | $\mathcal{L}_{space}$ | $\mathcal{L}_{output}$ | Kitchen | → | KITTI | Comic | → | Parasites |
|---|---|---|---|---|---|---|---|---|---|
| | | | | Old | New | Avg. | Old | New | Avg. |
| 1 | ✓ | × | × | 9.66 (-85.07) | 69.42 | 39.54 | 0.00 (-56.44) | 81.45 | 40.72 |
| 2 | × | ✓ | × | 31.85 (-62.88) | 69.53 | 50.69 | 17.36 (-39.08) | 79.09 | 48.22 |
| 3 | × | × | ✓ | 76.47 (-18.26) | 70.31 | 73.39 | 40.48 (-15.96) | 79.54 | 60.01 |
| 4 | ✓ | ✓ | × | 28.29 (-66.44) | 65.66 | 46.97 | 23.67 (-32.77) | 71.16 | 47.41 |
| 5 | ✓ | × | ✓ | 87.23 (- 7.50) | 71.09 | 79.16 | 47.88 (- 8.56) | 80.68 | 64.28 |
| 6 | × | ✓ | ✓ | 84.10 (-10.63) | 69.92 | 77.01 | 45.95 (-10.49) | 79.33 | 62.64 |
| 7 | ✓ | ✓ | ✓ | 89.70 (- 5.03) | 71.22 | **80.46** | 52.39 (- 4.05) | 81.07 | **66.73** |

exclusive reliance on either attention distillation or feature space distillation is insufficient to preserve comprehensive old knowledge. Student models trained with these individual approaches tend to overlook objects that appear only in old tasks, while the introduction of output distillation can further mitigate catastrophic forgetting. Achieving Pareto optimality for both new and old tasks is observed when combining all three distillation losses.

## 4.5 Scalability

We have demonstrated that PD effectively preserves old task knowledge on the one-stage detector YOLOv8 [10]. In this section, we

**Table 6: Results of implementation on other detectors. To validate the scalability of our method, we implement PD and competitive methods on the two-stage detector (Faster R-CNN) and the Transformer-based detector (DINO). Since CL-DETR is designed specifically for DETR and is hard to apply to other models, we only re-implemented it on DINO. Here CL-DETR* denotes we maintain a reservoir of exemplars as the original paper [22].**

| Detectors | Methods | Kitchen | → | KITTI | Comic | → | Parasites |
|---|---|---|---|---|---|---|---|
| | | Old | New | Avg. | Old | New | Avg. |
| Faster R-CNN (Two-stage model) | Teacher | 82.97 | - | - | 49.15 | - | - |
| | Finetuning | 0.00 (-82.97) | 68.26 | 34.13 | 0.00 (-49.15) | 72.34 | 36.17 |
| | Faster ILOD [26] | 78.55 (- 4.42) | 66.70 | 72.62 | 44.56 (- 4.59) | 72.76 | 58.66 |
| | ERD [5] | 74.04 (- 8.93) | 63.83 | 68.93 | 42.48 (- 6.67) | 70.11 | 56.29 |
| | Ours | 80.23 (- 2.74) | 67.12 | **73.67** | 47.96 (- 1.19) | 71.65 | **59.80** |
| DINO (Transformer) | Teacher | 95.72 | - | - | 60.80 | - | - |
| | Finetuning | 0.00 (-95.72) | 77.40 | 38.70 | 0.00 (-60.80) | 89.95 | 44.97 |
| | Faster ILOD [26] | 70.98 (-24.74) | 74.29 | 72.63 | 44.83 (-15.97) | 84.10 | 64.46 |
| | ERD [5] | 75.32 (-20.40) | 76.08 | 75.70 | 42.03 (-18.77) | 87.77 | 64.90 |
| | CL-DETR [22] | 12.04 (-83.68) | 76.48 | 44.26 | 3.36 (-57.44) | 90.03 | 46.69 |
| | CL-DETR* [22] | 75.73 (-19.99) | 78.15 | 76.94 | 38.95 (-21.85) | 88.52 | 63.73 |
| | Ours | 78.30 (-17.42) | 76.93 | **77.61** | 46.11 (-14.69) | 88.16 | **67.13** |

aim to explore the scalability of PD when applied to other detectors, the most classic two-stage detector Faster R-CNN [30] and a novel transformer-based detector DINO [37]. After modifying the detection heads, we re-implemented our method and compared it with Faster ILOD and ERD. For DINO, we additionally implemented the latest incremental learning method for transformer detectors, CL-DETR [22]. The results of incremental object detection are shown in Tab. 6. It is evident that despite not being specifically tailored for these models, PD also performs well on other types of detectors. CL-DETR, however, shows significant performance degradation. This occurs due to its use of pseudo-labeling on the current dataset to avoid catastrophic forgetting, which is not suitable when there is a significant disparity between the old and new datasets. We also observed a more severe forgetting when applying these CNN-based methods to DINO, with the drop of mAP on old tasks being nearly eight times higher compared to Faster R-CNN. We speculate this is due to the instability of the Hungarian matching [1] in the transformer detector, and we will focus on this issue in future work.

## 5 CONCLUSION

In this paper, we proposed Purified Distillation (PD) to mitigate catastrophic forgetting in incremental object detection with both domain shift and category gap. PD leveraged multi-scale cross attention distillation to adaptively focus on features relevant to old tasks and employed feature space distillation to guide students in learning the characteristics of both new and old tasks. Additionally, an entropy-guided selection strategy is introduced, integrating classification and location confidence to purify the most representative output nodes for output distillation. We conducted experiments across multiple scenarios using six datasets to simulate the complex demands of real-world applications. Extensive experiments underscored the generalizability and robustness of PD.

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
