# OpenReview forum: "Purified Distillation: Bridging Domain Shift and Category Gap in Incremental Object Detection"
_acmmm.org/ACMMM/2024/Conference — MM2024 Poster_

### Official Review · Reviewer_Y9WJ · 2024-05-11

**Rating:** 3
**Confidence:** 4

**Summary:**

This paper proposes a distillation-based incremental object detection method for handling both domain and class incremental learning. Multi-scale Cross Attention Distillation (MCAD), Feature Space Distillation (FSD), and Entropy Guided Output Distillation (EGOD) are designed to transfer knowledge from the teacher to the student. Single-step and multiple-step incremental learning experimental results on five datasets demonstrate the effectiveness of the proposed method.

**Strengths:**

1. It is a practical problem to handle both domain and class incremental learning simultaneously.
2. The proposed three types of distillation are interesting and effective.
3. Experiments are sufficient. The results are superior to the compared methods.

**Limitations:**

1. The authors’ insights into the unique problems of domain-incremental object detection should be presented in the introduction. Why previous incremental object detection methods can not be directly applied to this task? Why the proposed method can bridge domain shift and category gap simultaneously?
2. The reason why this method can eliminate dependence on exemplars from old tasks, as the author claimed, should be given theoretically. From the results of the implemented methods, these methods can also achieve satisfactory results without storing exemplars, such as Faster ILOD. This may conflict with the statement in the abstract: “Class IOD methods suffer from severe performance degradation in these scenarios with domain shifts.”
3. Several operations are designed for multi-scale features. Are there any experiments to verify the effectiveness of transferring multi-scale features rather than single-scale features? Previous methods are primarily designed for single-scale features.
4. How about the computational cost? The attention module for Multi-scale Cross Attention Distillation and convolutional layers for aligning feature maps both increase the parameters.
5. What’s the meaning of “We train the model on the old task from the official weights [10]” in Line 553?
6. The related works are not sufficient. Many recent works in 2023 and 2024 are not cited, such as:

a) Non-exemplar Domain Incremental Object Detection via Learning Domain Bias. AAAI 2024.

b) Domain Incremental Object Detection Based on Feature Space Topology Preserving Strategy. TCSVT 2024.

**Suitability:**

2

---

### Official Review · Reviewer_R4iW · 2024-05-22

**Rating:** 4
**Confidence:** 3

**Summary:**

The paper addresses the task of  incremental object detection with both domain shift and category gap.
The authors propose a novel purified distillation including multi-scale cross attention distillation, feature space distillation, and entropy guided output distillation. Extensive experiments across various Incremental Object Detection(IOD) settings on six datasets show that PD significantly outperforms state-of-the-art methods in preserving knowledge from previous tasks.

**Strengths:**

1. This paper is well-written and easy to follow with figure of overall framework.

2. The authors produce a novel purified distillation methods with state-of-the-art results.

   2.1 The authors accordingly bring discriminator to solve the gap between source domain and target domain in latent space

   2.2 They provide detailed explanations of the equations with figures as **fig. 3.**

3. These methods do not need extra data from source domain.(e.g. exemplars, expanded network)

4. This paper thoroughly demonstrates the effectiveness of the proposed method through experiments.

**Limitations:**

The paper demonstrate the effectiveness of the proposed **multi-scale cross attention distillation** method. Multi-scale distillation techniques are widely used in many computer vision tasks, however, also there are already existing papers that address **catastrophic forgetting** in incremental learning utilizing **additional learnable parameters(branch)**. (e.g. "FDCNet: Feature Drift Compensation Network for Class-Incremental Weakly Supervised Object Localization" and "Class-Incremental Learning with Strong Pre-trained Models")

**novelty and comparison:**

The authors need to describe the differences in **Sec. 2.** or **Sec. 3.**.

**Additional experiments:**

The ablation study on the three proposed losses is well conducted. However, if possible, I would like to see the experimental results varying with different $\lambda$ values.

**Suitability:**

2

---

### Official Review · Reviewer_RBKq · 2024-05-23

**Rating:** 4
**Confidence:** 3

**Summary:**

This work propose a novel knowledge distillation method to mitigate the catastrophic forgetting in incremental object detection. Three different distillation losses are proposed to address the domain shift and category gap. Extensive experiments demonstrate the effectiveness of the proposed methods.

**Strengths:**

- The experiments are solid.
- The idea of introducing the discriminator for distillation seems interesting

**Limitations:**

- In this paper, three losses are proposed. However, the motivation for using these three losses is unclear, and the authors are encouraged to give an explanation to show that the proposed method is not a simple combination of three losses.
- The latest baseline in Table2 is up to 2022, the authors are advised to compare with recent methods.
- The writing needs further improvement, and there are some grammatical errors.
- In Equation 3, the channel dimension has been global average pooled, why there still exists the channel dimension $d$?
- In the Multi-scale Cross Attention Distillation, the authors stated the query can obtain domain-specific feature by cross-attention, it needs more evidence to support this statement. Moreover, this Multi-scale Cross Attention Distillation is very similar to existing promptKD[1], with both using learnable parameters to query interesting region, except the additional multi-scale strategy used in this work. The author should clarify the specific difference.

> [1] PromptKD: Unsupervised Prompt Distillation for Vision-Language Models

**Suitability:**

2

---

### Meta-Review · Area_Chair_1MsK · 2024-07-02

**Recommendation:** Accept (Poster)
**Confidence:** 4

**Metareview:**

This paper proposes a distillation-based incremental object detection method for handling both domain and class incremental learning. All reviewers agreed to accept this paper after the rebuttal period. Therefore, my final recommendation is accept (poster).